# A sub-wavelength Si LED integrated in a CMOS platform

Zheng Li [1,4] ✉, Jin Xue [1,3,4], Marc de Cea [1], Jaehwan Kim[1], Hao Nong[2], Daniel Chong[2], Khee Yong Lim[2], Elgin Quek [2] & Rajeev J. Ram [1] ✉

A nanoscale on-chip light source with high intensity is desired for various applications in integrated photonics systems. However, it is challenging to realize such an emitter using materials and fabrication processes compatible with the standard integrated circuit technology. In this letter, we report an electrically driven Si light-emitting diode with sub-wavelength emission area fabricated in an open-foundry microelectronics complementary metal-oxide-semiconductor platform. The light-emitting diode emission spectrum is centered around 1100 nm and the emission area is smaller than 0.14 μm² (~ ⌀ 400 nm). This light-emitting diode has high spatial intensity of >50 mW/cm² which is comparable with state-of-the-art Si-based emitters with much larger emission areas. Due to sub-wavelength confinement, the emission exhibits a high degree of spatial coherence, which is demonstrated by incorporating the light-emitting diode into a compact lensless in-line holographic microscope. This centimeter-scale, all-silicon microscope utilizes a single emitter to simultaneously illuminate ~9.5 million pixels of a complementary metal-oxide-semiconductor imager.

In the past two decades, various photonic components, such as couplers, waveguides, and modulators, have been successfully integrated into microelectronics platforms, which enables researchers to translate various bench-top optical systems to photonic chips[1]. Some examples of applications include data communication[2,3], sensing[4], imaging[5] and quantum computing[6]. Despite this progress, a small and bright on-chip emitter remains elusive and thus in most photonic chips the light originates from off-chip light sources, which leads to low overall energy efficiency and fundamentally limits the scalability of photonic chips[7,8]. Driven by the desire for higher integration density, researchers have fabricated on-chip emitters using various material systems such as rare-earth-doped glass[9,10], Ge-on-Si[11,12], and heterogeneously integrated III–V materials[8]. Emitters based on these materials can have good device performance but the associated fabrication processes are still challenging to integrate into standard complementary metal-oxide-semiconductor (CMOS) platforms[13]. Monolithically integrated native Si is also a promising candidate material for

nanoscale and individually controllable emitters due to the nanometer fabrication precision and the large integration scale. However, Si emitters suffer from low quantum efficiency because of the indirect bandgap. This fundamental disadvantage combined with the limitations set by the available materials and fabrication tools hinder the realization of a small native Si emitter in CMOS.

Tremendous effort has been made to enhance the light emission of Si. Some early work by Green et al. focused on using large-area, high-quality single crystalline Si with dedicated photon extraction structures[14,15]. These emitters are several cm² in size and have intensities of $10^{-1}$ mW/cm². These large-area emitters rely on the low surface-to-volume ratio to minimize the non-radiative recombination on the surfaces, and thus it is challenging to reduce the emission area while keeping high efficiency. Other strategies include carrier confinement[16–23], defect activation[17,24–26], field emission effect[27], Purcell enhancement[23,26,28], and avalanche effect[29–31]. To date, the Si emitter with the smallest emission area (≈1 μm²), as well as the highest intensity

[1]Research Laboratory of Electronics, Massachusetts Institute of Technology, Cambridge, MA 02139, USA. [2]GlobalFoundries Singapore Pte. Ltd., Singapore 738406, Singapore. [3]Present address: Institute of Microelectronics (IME), A*STAR, Singapore 138634, Singapore. [4]These authors contributed equally: Zheng Li, Jin Xue. ✉e-mail: zhli@mit.edu; rajeev@mit.edu

($\approx$600 mW/cm$^2$), is reported by Schmitt et al.[23]. In their emitter, the carrier recombination is confined in an inverse tapered Si half-ellipsoidal nanostructure which is also an optical cavity. However, their emitter requires an AFM top contact, which hinders the direct compatibility with CMOS platforms. The smallest emitter fabricated in CMOS was reported in 2021, and employed vertical pn junctions in an unmodified, open-foundry microelectronic CMOS node[32]. These devices were scaled to an emission area of 4 $\mu$m diameter and intensity of over 40 mW/cm$^2$.

To further scale the emission area down to sub-micrometer dimensions in CMOS platforms, several issues need to be addressed. First, although bimolecular recombination favors carrier confinement[33], as the surface-to-volume ratio of the active region increases, non-radiative recombination assisted by surface defects (Shockley-Read-Hall recombination, SRH) becomes significant. This is a common problem in most nanoscale opto-electronic devices and might be partially solved by surface passivation[34]. Second, heat dissipation needs to be efficient to prevent thermal droop of quantum efficiency when the current density in the active region is high. As has been reported in Si nano-crystal-based emitters, efficiency droop happens at current densities as low as 1 $\mu$A/cm$^2$ and limits the intensity below 1 mW/cm$^{2[18,19]}$. Third, the carrier injection structure has to be optimized to support high current while not perturbing light extraction. In our previous work, we have shown that vertical pn junctions with top metal contacts can support high injection current while reducing device footprints compared to lateral junctions[32]. However, the opaque metal contact leads to significant shadowing if the emission area shrinks to a comparable size.

In this work, we report an electrically-driven, nanoscale Si light-emitting diode (LED) realized in an unmodified, open-foundry microelectronic CMOS node. At room temperature (-22 °C), our LED exhibits a sub-wavelength emission area (<0.14 $\mu$m$^2$, or $\varnothing$ 400 nm) and high spatial intensity (>50 mW/cm$^2$) with the emission spectrum centered around 1100 nm. No efficiency droop is observed at current density up to 2 MA/cm$^2$. The aforementioned side effects with scaling are mitigated as follows. First, the carriers are spatially confined by the gate oxide layer and a strong electrical field introduced by the nanoscale top contact. This confinement configuration does not rely on extra interfaces except the well-passivated gate oxide. Second, carrier recombination happens close to the Si substrate which conducts heat effectively. Moreover, inspired by the reports on the emission associated with gate oxide breakdown[35,36], the top contact is fabricated using gate poly-Si instead of metal. The contact is thus transparent and supports further scaling.

Due to sub-wavelength confinement, the emission is spatially coherent. The aperture of the device serves as its own spatial filter. As a potential application, a lensless digital holographic microscope was built using the LED. We demonstrate that a single LED is bright enough to simultaneously illuminate -9.5 million pixels (>1 cm$^2$ area) of a Si CMOS camera and that the signal-to-noise ratio (SNR) of the holograms is sufficient to reconstruct the images of randomly distributed 20 $\mu$m diameter latex beads.

## Results
### Device structure and characterization
Our LED is fabricated in a 55 nm CMOS node, alongside other functional photonic and electronic components all integrated on the same chip (Fig. 1(a)). In Fig. 1(b) (c), we present the schematic of our LED. The LED denoted in the right dashed box in Fig. 1(b) is a vertical n+/n/p junction. Here the n+, n and p refer to the gate poly-Si layer (degenerate n-doped), the n-well (intermediate n-doped) and the substrate (lightly p-doped), respectively. The thin (-100 nm) gate poly-Si serves as a transparent top contact to the device. The n-well is -1.6 $\mu$m in $x$, -1.8 $\mu$m in $y$ and < 2 $\mu$m in $z$. Only a small area (-0.3 $\mu$m$^2$) of the n-well is in contact with the gate oxide (<3 nm thick). This

opening is defined by shallow trench insulator (STI). In this area, a Si filament is formed in the gate oxide by applying electrical stress. This process is usually referred to as hard breakdown[37,38]. Microscopically, when high-voltage gate bias is applied, traps form in the gate oxide and a conduction path is created. Near this path, the gate current leads to thermal damage which creates more traps. These newly created traps merge into the conduction path which further increases the conductance. This thermal runaway process leads to a point when the Si-O bonds break and the Si in the conduction path melts to form a Si filament. The cross-section of the breakdown site is usually sub-100 nm and the Si filament is a resistor in series with the gate poly-Si and the n-well[38]. Similar methods were used to fabricate contacts from gate poly-Si to Si substrates by refs. [35,36]. In our test, the substrate was biased at 6.0 V with the poly-Si grounded to introduce gate oxide breakdown. We observed that the current rapidly increased from pA to mA and was stabilized within 1 min. After the breakdown, the IV characteristic was stable and showed no obvious hysteresis.

The carrier transport and recombination process in the LED are illustrated in Fig. 1(c) in which the gate poly-Si is grounded and the substrate is at positive bias. Due to the negative gate bias, a hole accumulation layer forms near the gate oxide. In the accumulation layer, the holes recombine near the Si filament where electrons are injected from the poly-Si[35]. This process leads to a highly localized emission spot. As the bias increases, the holes are attracted and confined by the increasing electrical field in the Si filament and the emission becomes more localized. In Fig. 1(d), a micrograph with the LED turned on is presented and the bright spot is observed. As the electrons diffuse into the substrate, they recombine with the majority holes and the associated emission is spatially broad because the electron diffusion length can be several millimeters in the lightly p-doped substrate[39]. The radiative recombination of holes and the majority electrons in the n-well bulk also contribute to the total emission but is insignificant since the size of the n-well (<2 $\mu$m) is much smaller than the diffusion length of the minority holes (10–100 $\mu$m in intermediate n-doped Si)[40]. Therefore, the emission in our LED mainly consists of two spatially separate components – a bright, localized spot and a faint, spatially broad background.

The emission mechanism is verified by measuring the emission spectra of the LED and a reference emitter which is fabricated close to the LED in the identical geometry and with similar doping levels but with p-poly-Si top contact and a p-well. The results are presented in Fig. 1(e), where the emission was coupled into a single-mode fiber (SMF) to spatially filter the background. The microscope and the spectrometer used for the characterization are presented in Methods and Supplementary Section 1. Although the reference emitter also emits photons, the emission spectrum is distinct from the LED. The LED spectrum peak is around 1100 nm, which is associated with the phonon-assisted band-to-band transition. The spectral full-width-half-maximum (FWHM) of the emission increases monotonically from -150 nm to -190 nm (Fig. 1(e) inset). Compared with Si LEDs on micrometer scale, our emission spectrum is 50–150% broader. This is in agreement with the reports on the emission from gate oxide breakdown[35,36,41–43]. The broadening can be attributed to the elevated temperature of the hot carriers accelerated by the electrical field in the Si filament. As the bias increases, the carrier temperature increases and thus the spectral FWHM of the emission also increases. However, the emission peak around 1100 nm shows no red-shift with increasing bias while the red-shift with increasing temperature is observed in other Si LEDs because of bandgap shrinking[14,16]. This phenomenon indicates that the lattice temperature of our LED does not increase significantly because the Si substrate dissipates heat efficiently. This is elaborated in Supplementary Section 6 where the heat dissipation from the active region to the substrate is simulated and compared with alternative device configurations. As for

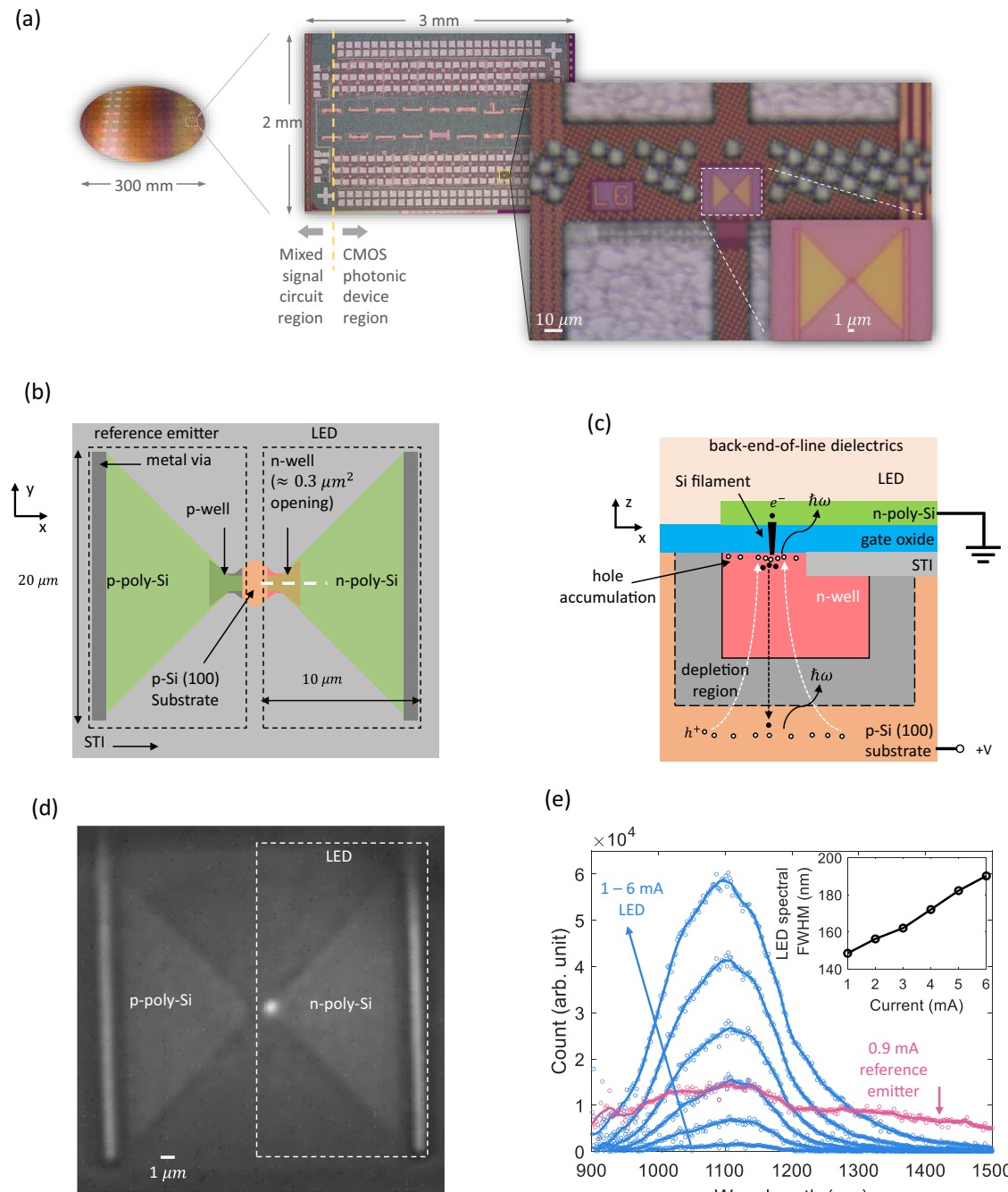

**Fig. 1 | Device structure and emission spectra. a** Photograph of a fully fabricated 300 mm wafer with monolithic electronics and photonics, and optical micrograph of a diced, unpackaged chip with different active and passive photonic components and mixed-signal circuits integrated side-by-side, and close-ups of the LED and the reference emitter on this chip. **b** A schematic top view of the LED and the reference emitter. Here the back-end-of-line (BEOL) dielectrics and the gate oxide are not shown. STI: shallow trench insulator. **c** A zoom-in side view of the LED on the white dashed line in **b** and the corresponding carrier transport. The solid and the hollow circles indicate electrons and holes, respectively. The black and the white dashed arrows indicate electron and hole transport, respectively. **d** A micrograph of the LED when it is biased at 6 mA. The wide-field illumination light is from a commercial LED centered around 1100 nm. **e** Spectra of the LED and the reference emitter were measured by routing the emission through a single-mode fiber into a spectrometer based on an InGaAs camera. (Supplementary Section 1.) The hollow circles are the raw data and the solid lines are from the Savitzky–Golay filter with polynomial order 3 and frame length 21. The spectral full-width-half-maximum (FWHM) of the LED is presented in the inset.

the reference emitter, although the peak around 1100 nm is observable, the spectrum is much broader than those of the LED. Since the reference emitter (p+/p/p) is uni-polar, the light emission is mainly from impact ionization and hot carrier transition as discussed by refs. [36,42]. The comparison of the distinct spectra confirms that phonon-assisted band-to-band transition dominates the radiative recombination in the LED while intraband transition and

hot-carrier-induced transition involving higher energy bands are not significant. The broad spectrum of the reference emitter also serves to validate the broad optical bandwidth of the collection optics and the detectors.

Images of the LED emission at various currents are presented in Fig. 2(a)−(c). The focus position was adjusted such that the signal at the center of the emission was maximized. The emission pattern was

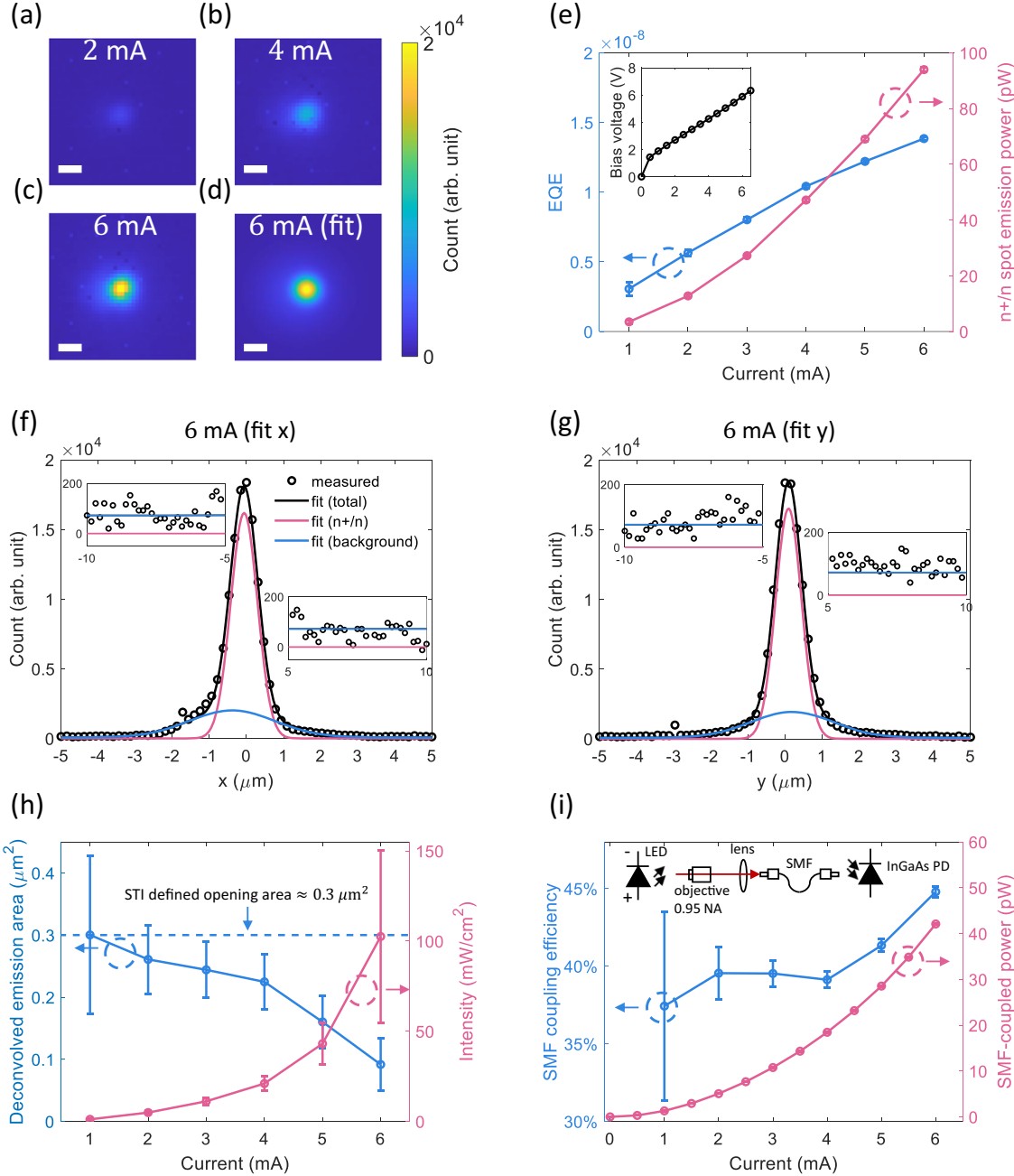

**Fig. 2 | Characterization of the LED. a–c** Images of the emission pattern at multiple currents with 50 ms integration. A microscope equipped with a 100×, 0.95NA objective and an InGaAs camera were used to characterize the devices. (Supplementary Section 1). The chip die was wirebonded to a chip carrier and the carrier was fixed on a piezo translation stage. Scale bar: 1 μm. **d** 2D Gaussian fit of the emission pattern at 6 mA. Scale bar: 1 μm. **e** n+/n emission power and the associated external quantum efficiency (EQE) with the background emission neglected. The inset shows the forward bias voltage versus current. The error bars are from the 95% confidence interval of the corresponding fitting parameters. **f, g** x and y cross-sections of the fit emission pattern at 6 mA from −5 to 5 μm. The inset figures are fit from ±5 to ±10 μm. Only the 20 × 20 μm² area centered at the emission spot is considered in the fit because the emission outside of this region is shadowed by metal fill required by the process. **h** Deconvolved emission area and spatial intensity. The deconvolution results are based on the spatial full-width-half-maximum (FWHM) of the point-spread function (PSF) of the microscope. (784 ± 50 nm in x and 740 ± 46 in y. Methods and Supplementary Section 2.) The dashed line indicates the shallow trench insulator (STI) defined opening area which confines carriers when the bias is low. **i** Single-mode fiber (SMF) coupled power and coupling efficiency. The schematic setup is presented to indicate the measurement methods. The SMF is PM980 (Thorlabs). The error bars in **h** and **i** are from the error propagation considering the PSF measurement error and the fit error in **e**.

characterized by fitting the images into two 2D Gaussian functions. (Methods and Supplementary Section 4.) As an example, the 2D fit result of 6 mA is presented in Fig. 2(d). The components of the fit in the x and y cross-sections are presented in Fig. 2(f) and (g), respectively. The fit results are in good agreement with the measured data and clearly consist of the two expected components of the emission. At the center of the LED, a bright spot dominates the intensity, while the diffusive background extends ±10 μm away from the center (Fig. 2(f) (g) inset). This bright spot is referred to as the n+/n emission spot in the following text since the corresponding carrier recombination happens near the n+/n interface.

In Fig. 2(e), we present the n+/n emission power and the external quantum efficiency (EQE). The emission power was evaluated by integrating the Gaussian fit of the spot corrected by the sensitivity of

the camera and the optical power transmission of the microscope. (Methods and Supplementary Sections 3, 4.) At 6 mA, the n+/n emission power and the associated EQE are ~94.0 pW and ~$1.4 \times 10^{-8}$, respectively. The monotonically increasing EQE with current indicates a superlinear dependence of the n+/n emission power on the injection current, which is a result of carrier confinement because the bimolecular recombination rate scales with the product of the excess carrier concentrations. Note that the EQE shows no thermal droop even when the current density is above 2 MA/cm$^2$ (a conservative estimation using the 0.3 μm$^2$ opening area as the cross-section). This is mainly because the active region is embedded in the Si substrate which is an efficient heat sink. (Supplementary Section 6.)

In Fig. 2(h) we present the emission area and the spatial intensity of the n+/n emission spot. The emission area is estimated by deconvolving the spatial FWHMs of the spot from the Gaussian fit using the independently measured point-spread function (PSF) of the microscope. (Methods and Supplementary Sections 2, 4.) From 1 to 6 mA, the emission area shrinks from $0.30 \pm 0.12$ μm$^2$ to $0.09 \pm 0.04$ μm$^2$. This is consistent with the carrier transport process in which the holes are confined by the electrical field in the Si filament and the localization becomes stronger with increasing bias. Below 4 mA, the emission area is close to the STI-defined opening area (Fig. 1(b),(c)). This indicates that at low bias the holes are mainly confined in the n-well by the STI and thus the emission area does not decrease significantly. The spatial intensity is evaluated using the emission area and the power of the n+/n emission spot (Fig. 2(e)). At 6 mA, the intensity is $102 \pm 48$ mW/cm$^2$. Compared with the smallest CMOS emitters reported previously[32], the emission area is 2 orders of magnitude smaller and the average intensity is approximately doubled. The improvement of our LED is achieved first by using transparent top contact to prevent shadowing and enhance light extraction. Also, the top contact and the active region are not directly in contact, but through the sub-100 nm Si filament formed in the gate oxide. This configuration spatially confines holes in all x-, y-, and z-directions, which reduces the emission area and enhances the spatial intensity. In contrast, in ref. [32], the top contact is directly fabricated on the bulk active region and the carriers can diffuse several to tens of micrometers laterally. Moreover, SRH recombination is minimized because the active region is away from surfaces except the well-passivated gate oxide while in ref. [32] multiple interfaces exist in the active region and a higher SRH recombination rate is therefore expected.

The compactness of the emission is further verified by measuring the optical power coupled into an SMF. In Fig. 2(i), we present the SMF-coupled power and the SMF coupling efficiency versus the injection current corrected by the optical power transmission of the microscope. (Supplementary Section 3.) The SMF-coupled power was maximized by moving the sample using a piezo stage. During the test, the *xyz* position corresponding to the maximum power did not change with current. At 6 mA, the SMF-coupled power is ~42 pW. The SMF coupling efficiency is computed by dividing the SMF-coupled power by the n+/n emission power (Fig. 2(e)). From 4 mA to 6 mA, the SMF coupling efficiency increases from ~40 to ~45% while it stays ~40% under 4 mA. The high SMF coupling efficiency is due to the sub-wavelength emission area and the trend is consistent with the decrease of the emission area above 4 mA (Fig. 2(h)).

Our LED is the smallest reported Si emitter and its intensity is comparable to state-of-the-art Si emitters with much larger emission areas. A detailed benchmark of our LED compared with other reported emitters is presented in Supplementary Section 5. The results presented above are all at room temperature and at steady state. In Supplementary Section 7, we present the device performance at elevated temperatures. We observe that the SMF-coupled power increases with temperature from 10 °C to 70 °C. In Supplementary Section 9, we present the time-resolved optical power of the LED and demonstrate a switching bandwidth of 77 MHz.

The performance variance of our LEDs on multiple chip die is modest. In Supplementary Section 8, we present the SMF-coupled powers and the bias voltages of five devices on different chip die. The relative (absolute) standard deviations of the SMF-coupled powers and the bias voltages are ~13.7% (5.0 pW) and ~5.3% (0.34 V), respectively, at 6 mA. The power of the worst device is ~80% of the average device. These preliminary results indicate good reproducibility of our devices. The LEDs are also reliable over time if operated below 7 mA. After being turned on and off for ~$10^5$ times as a reliability test, the SMF-coupled power decreases by ~25% and no significant optical power decrease is observed after the test. (Supplementary Section 8.) This is probably associated with lateral propagation of breakdown sites[37]. The device becomes less reliable at high injection. With 7 mA injection, the optical power decreases by more than 50% within a minute. To be conservative, all the results presented in this report are from the LEDs which were slightly degraded during the reliability test.

We also characterized the reference emitter and the results are presented in Supplementary Section 10. In summary, compared with the LED, besides the distinct emission spectra presented in Fig. 1(e), the reference emitter has much larger resistance (>20 kΩ) and is less reliable even at low current injection (~0.9 mA). We conclude that the reference emitter is not suitable for an on-chip light source because of the requirement for high bias voltage and the lack of reliability.

## Digital in-line holography

The small emission area of our LED results in highly spatially coherent light which enables interference-based applications including holographic microscopy. We incorporated the LED as a point source for direct illumination in a lensless in-line holography setup (Fig. 3(a)), by which an interference pattern between the unscattered incident light and the light scattered by a sample was generated and recorded with a Si CMOS camera of 1.76 cm × 1.33 cm size. We used this setup to image a sample containing 20 μm diameter latex beads randomly distributed on a glass slide. In Fig. 3(b), (c), and (d), we present the full hologram on the imager, a close-up of the hologram and the digital counts on a line segment, respectively. In Fig. 3(c) (d), the zeroth order bright spots and the first order bright rings are observed, which verifies that the spatial coherence of our LED is high enough to compensate for the relatively broad spectrum to generate interference patterns. In Fig. 3(b), we denote a white dashed box of ~1.25 cm × 1.10 cm, where the interference pattern of the beads can be clearly observed. This illumination area corresponds to a numerical aperture (NA) of NA$_x \approx 0.46$ and NA$_y \approx 0.41$. The high NA of our LED (by virtue of its small emission area) allows us to achieve a large field of view with minimal distance between the source and the sample (1 cm). In conventional in-line holography setups employing LEDs and pinholes such a distance is on the order of 10 cm or above because the illumination NA is limited to $\approx 0.62 \lambda/r$ for a circular pinhole with tens μm diameter ($\lambda$ is the illumination wavelength and $r$ is the radius of the pinhole.) The pinhole-free configuration of our setup also simplifies the alignment.

We performed hologram reconstruction using the angular spectrum method[44]—the recorded hologram was transformed to the frequency domain through fast Fourier transform (FFT), back-propagated to the image plane in the frequency domain and then converted back to the space domain with inverse FFT. The reconstruction result is presented in Fig. 3(e) (f). Compared with the optical micrograph (Fig. 3(g)), the clusters of the beads can be clearly resolved from each other and their relative positions are accurate. The reconstruction also exhibits the details of the sample. In Fig. 3(f) (g), three clusters containing 1, 2, and 3 beads are labeled. Their shapes and sizes are distinct. We thus conclude that the resolution of our holographic microscope is approximately the size of 1 bead (~20 μm). The limitations of this simple reconstruction algorithm include the assumption

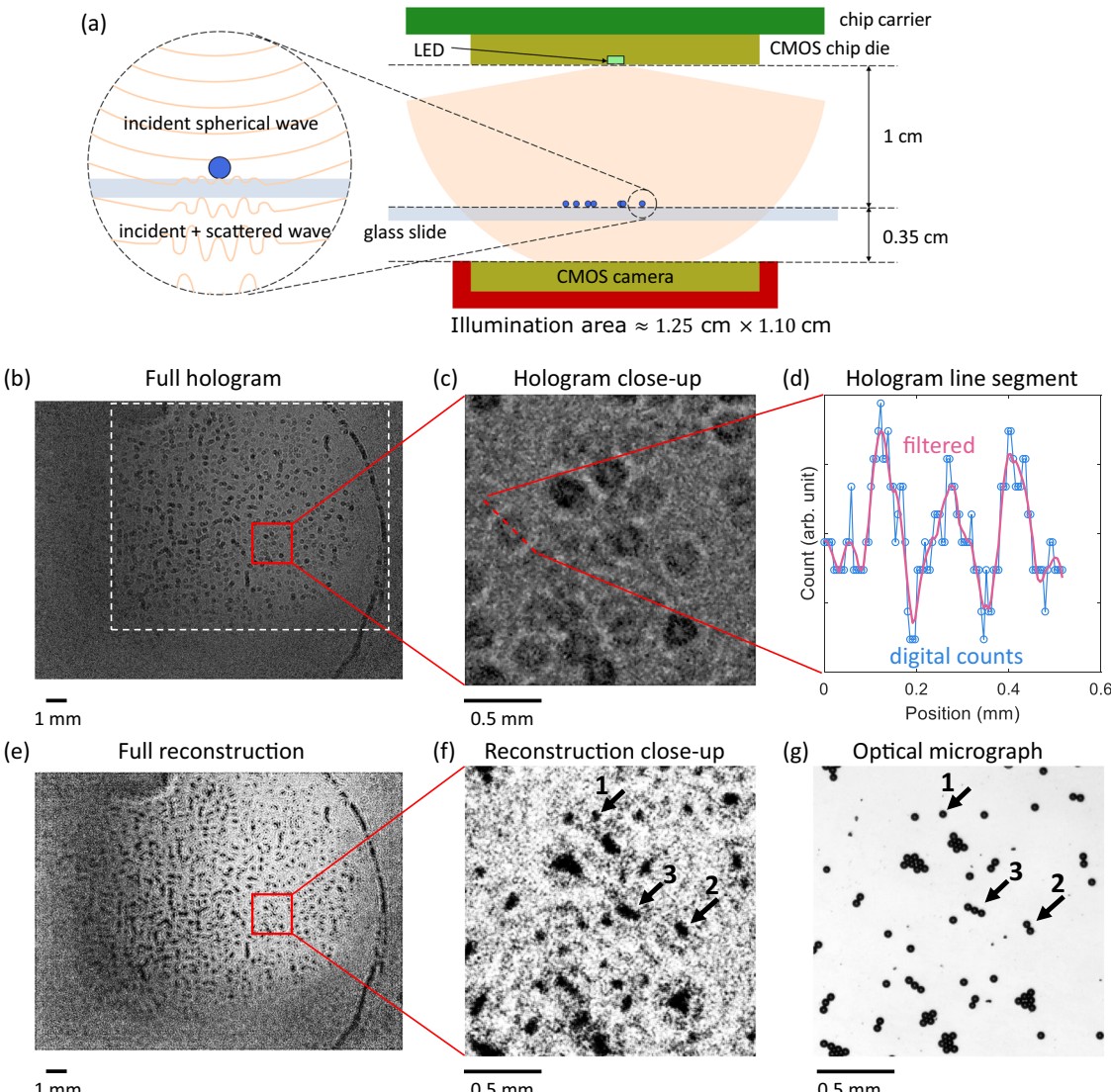

**Fig. 3 | Digital in-line holography setup and results. a** Schematic of the experimental setup demonstrating in-line holographic imaging of 20 μm diameter latex beads with a single LED (bias current at 6.5 mA) as the illumination source. Close-up on the left illustrates the interference pattern formed between the incident light and the scattered light from a bead. **b** Full hologram recorded by a cooled CMOS camera (5 °C) for 16 s integration. The imager chip size and the pixel pitch are 1.76 cm × 1.33 cm and 3.8 μm, respectively. The raw hologram is filtered by a 5 × 5 median filter. The white dashed box indicates the illumination area. **c** Close-up of the hologram in the red box in **b**. **d** Digital counts of the hologram on the red dashed line in **c**. The filtered curve is from Savitzky–Golay filter with polynomial order 3 and frame length 11. **e** Reconstruction of the hologram on the whole imager. The contrast is enhanced by histogram equalization with 64 bins. **f** Close-up of the reconstruction in the same area as **c**. **g** Optical micrograph of the sample in the same area as **c** and **f**, taken by a 5× objective.

of a monochromatic source (which is not true in our case), and the presence of the twin image. Such limitations can be overcome by employing more advanced techniques such as iterative phase retrieval[45,46] and machine learning-based reconstruction methods[47]. In our case, machine learning approaches robust to low SNR[48] are particularly attractive.

## Discussion

In this report, we present a CMOS-integrated sub-wavelength scale LED at room temperature exhibiting high spatial intensity ($102 \pm 48$ mW/cm²) and the smallest emission area ($0.09 \pm 0.04$ μm²) among the Si emitters in the literature. The maximum optical powers extracted from a single LED are ~94 pW in free space and ~42 pW in a SMF at 6 mA injection. In our LED, the heat is dissipated efficiently through the Si substrate, so no thermal droop of the EQE is observed even when the current density is above 2 MA/cm². The top contact is designed such that a strong electrical field spatially confines the

carriers and the only surface that contributes to the confinement at high bias is the well-passivated gate oxide. The top contact is also transparent to avoid shadowing on the emission, which supports further scaling. The device performance can be potentially improved by several modifications. First, the top contact should be optimized to support higher electron injection given that both the optical power and the EQE increase with current. This can be achieved, for example, by replacing sharp corners of the poly-Si contact by rounded ones to lower local electrical fields and prevent lateral propagation of breakdown sites. It is also possible to silicide part of the contact taper to reduce the parasitic voltage drop. Second, shallow junctions can be fabricated close to the active region to provide stronger electron confinement. Moreover, the p-substrate contact, which is ~2 mm away from the active region, can be fabricated closer to the n contact. This can lower the background emission and increase the bandwidth.

As a potential application, we integrated the LED into an in-line holographic microscope. The large NA and the high spatial coherence

of our LED combined brought us a centimeter-scale, all-silicon holographic microscope requiring no lens or pinhole. These LEDs can also be arrayed in CMOS to generate programmable coherent illumination for more complex systems in the future. While a simple FFT-based algorithm was used to reconstruct the image, we obtained reasonably good quality results. More advanced computational techniques can be used to improve the reconstruction. Besides the demonstrated application in holography, the presented LED is potentially useful in multiple other scenarios. For example, since the wavelength is within the minimum absorption window of biological tissues[49], together with its high intensity and nanoscale emission area, the LED can be ideal for bio-imaging and bio-sensing applications, including near-field microscopy and implantable CMOS devices. Also, it is possible to integrate the LED with on-chip photodetectors and the LED can then find its applications in on-chip communication, NIR proximity sensing, and on-wafer testing of photonics.

## Methods

### Device fabrication

The devices reported here were fabricated by GlobalFoundries, Singapore using a standard open-foundry CMOS technology (55BCDLite). 55BCDLite is a 12-in. mature process that has been widely used in various commercial products including analog amplifiers, digital signal processors, and power devices. Our device design and layout utilized standard mask layer sets and complied with critical design rules, demonstrating its feasibility of monolithic integration with a variety of microelectronics to realize more complex systems.

### Characterization methods

A microscope was built to characterize the LED and the reference emitter. A schematic of the microscope is presented in Supplementary Fig. 1. The emission of the LED was collected by a 0.95 NA objective and was either spatially resolved as a wide-field image using an InGaAs short-wavelength infrared (SWIR) camera (Photonic Science) or coupled into a single-mode fiber (SMF, PM980, Thorlabs). Through the SMF, the emission was delivered to an InGaAs photodiode (PD, HP) to measure the SMF-coupled optical power or a dispersive spectrometer to measure the emission spectra. The spectrometer was based on the same SWIR camera and was wavelength-calibrated with a krypton lamp (Ocean Optics) and intensity-calibrated with a 2800 K blackbody source (Thorlabs). The chip die was wirebonded to a chip carrier and biased using a source meter unit (SMU, Keysight). The chip carrier was fixed to a piezo translation stage (Physik Instrumente).

The PSF of the microscope is $784 \pm 50$ nm in $x$ and $740 \pm 46$ nm in $y$. The PSF was characterized by imaging a target with sharp edges and fitting the image profiles into erf functions. Two typical fit results and the associated images are presented in Supplementary Fig. 2. We used 1100 nm wide-field illumination during the PSF calibration to match the emission of our LED. The PSF was used to deconvolve the emission pattern and to estimate the spatial intensity (Supplementary Section 4).

The optical power transmission of the microscope is ~53% on average near 1100 nm and is polarization-insensitive (50.1% and 56.2%, respectively with $x$- and $y$-polarization), which was calibrated by back-propagating amplified spontaneous emission (ASE, centered around 1130 nm) of a semiconductor optical amplifier (SOA, Innolume GmbH) and measuring the optical power after the objective. During the calibration, an iris was used to keep the ASE beam size the same as the back aperture of the objective. A polarizer and a half-wave plate were used to control the ASE polarization. More experimental details are discussed in Supplementary Section 3.

The image of the emission pattern was fit into two 2-D Gaussian functions which represent the n+/n emission spot and the background respectively. The power of the n+/n emission spot was estimated by integrating the digital counts of the corresponding fit profile. The integrated digital count was converted to optical power by considering the camera sensitivity ($\approx 19.56$ photo-electrons per digital count) and its quantum efficiency ($\approx 75\%$ around 1100 nm). The size of the spot was then estimated by deconvolving the fit profile with the calibrated PSF. The dependence of the spot size on the injection current is presented in Supplementary Fig. 3. More discussions on the fit and deconvolution methods can be seen in Supplementary Sections 3 and 4.

We modified the microscope to perform experiments at elevated temperatures. (Supplementary Section 7.) An aluminum heat sink in contact with a thermoelectric cooler and a feedback thermistor were installed between the piezo stage and the chip carrier. The temperature of the heat sink could be set from 5 °C to 85 °C with a precision better than $0.5$ °C using a PID microcontroller. During the measurements of the SMF-coupled power, once the temperature was changed and stabilized, the $z$-position of the piezo stage was tuned by a few micrometers to compensate for the thermal expansion of the heat sink and to maximize the coupled power.

To measure time-resolved optical power and switching bandwidth (Supplementary Section 9), we used an arbitrary waveform generator (AWG, Agilent Technologies) to bias the LED. The time-resolved optical power was measured using a time-correlated single-photon counting (TCSPC) system which utilizes a single photon avalanche diode (SPAD, PerkinElmer) and timing electronics (Becker and Hickl GmbH). The 3-dB switching bandwidth was estimated by measuring the frequency-dependent average optical power of the LED under 50% duty cycle square wave bias modulation.

## Data availability

The data that support the findings of this study are available within the letter and its supplementary information. All other relevant data are available from the corresponding authors upon request.

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

## Acknowledgements

This research is supported by the National Research Foundation (NRF), Prime Minister's Office, Singapore under its Campus for Research Excellence and Technological Enterprise (CREATE) program. Disruptive Sustainable Technology for Agricultural Precision (DiSTAP) and Critical Analytics for Manufacturing Personalized-Medicine (CAMP) are interdisciplinary research groups (IRG) of the Singapore-MIT Alliance for Research and Technology (SMART) Centre.

## Author contributions

Z.L. proposed the reported operation mode of the device, built the characterization tools, characterized the device, and drafted the manuscript. J.X. designed the device and performed initial tests in a different operation mode reported elsewhere. M.C. built the digital in-line holography setup and performed the image reconstruction. J.K. assisted J.X. with the tape-out design and the initial device tests. H.N., D.C., K.Y.L., and E.Q. fabricated the chip. R.R. supervised the project. All authors contributed to the final manuscript.

## Competing interests

The authors declare no competing interests.
