## [Peer Review File · Nature Communications]

A sub-wavelength Si LED integrated in a CMOS platformREVIEWER COMMENTS

Reviewer #1 (Remarks to the Author):

Summary:

The authors present a nanoscale LED whose fabrication is claimed to be CMOS compatible. The device has an extremely small emission area (the smallest every reported in Silicon based LEDs) achieved by the formation of a silicon filament for efficiency current injection. Results are presented in terms of electrical characteristics, external quantum efficiency, output optical power and spectral response. As a potential application, the device is used in digital holography, and successfully demonstrate the capability to produce interference (therefore certain degree of light coherence). The manuscript is highly interesting and novel results are achieved, therefore I suggest publication if the authors address the following points:

-It is mentioned several times that there is efficient heat sink due to the Silicon substrate. However, the heat spreading from the diode to the silicon substrate is not clear. Please add the clarification of the heat spreading mechanisms and potential limitations.

-The authors should specify the limits on current injection and highest output power that is possible to extract out of the LED.

-Unless I overlooked it, the manuscript does not specify at which temperature were the measurements carried out. Furthermore, it would be highly complementary to measure the light emission/EQE at different temperatures. The authors must consider additional measurements at different temperatures to have additional insights on the physical mechanisms limiting efficiency.

-The authors should mention the potential modifications of the structure to achieve higher optical output powers.

-The authors mentioned that the LED structures on multiple chips do not vary significantly. This variance should be quantified to give insights on the reproducibility of the fabrication process (especially the reproducibility of the silicon filament formation is a concern to me).

Reviewer #2 (Remarks to the Author):

Nanoscale on-chip light sources with high intensity are desired for various applications in integrated photonics systems. This paper reported an electrically driven Si light-emitting diode (LED) with sub-wavelength emission area fabricated in an open-foundry microelectronics CMOS platform. The emission area is smaller than $0.14 \mu\text{m}^2$ and this LED has high spatial intensity of $> 50 \text{ mW/cm}^2$, while the LED

emission spectrum is centered around 1100 nm. This work is very impressive and it can be accepted if the authors address the following comments.

1. The authors mentioned "In our previous work, we have shown that vertical pn junctions with top metal contact can support high injection current while also lower the device footprint compared to lateral junctions.³² However, the opaque metal contact leads to significant shadowing if the emission area shrinks to a comparable size. ... Compared with the work on the smallest CMOS emitters,³² the emission area is 2 orders of magnitude smaller and the average intensity is approximately doubled". From this description, the difference between the present work and the previous work in 32 is that the top contact is replaced by a n-poly-Si. Is there any other significant difference? Please clarify more details about this. What is the major reason for the 2 orders of magnitude improvement?
2. The present LED emits with the emission spectrum centered around 1100 nm. Please tell a little more details about the potentials the present LED for various applications.
3. Is the wavelength tunable?
4. The authors described the formation of the Si filament whose size is about usually sub-100 nm. Is the position of the Si filament controllable, or is it a kind of random? Is there any influence on the device performance?
5. Could this micro-LED be modulated fast? It will be very nice if the authors can give a comment and explanation on this.

To Reviewer 1:

The authors present a nanoscale LED whose fabrication is claimed to be CMOS compatible. The device has an extremely small emission area (the smallest every reported in Silicon based LEDs) achieved by the formation of a silicon filament for efficiency current injection. Results are presented in terms of electrical characteristics, external quantum efficiency, output optical power and spectral response. As a potential application, the device is used in digital holography, and successfully demonstrate the capability to produce interference (therefore certain degree of light coherence). The manuscript is highly interesting and novel results are achieved, therefore I suggest publication if the authors address the following points:

1) It is mentioned several times that there is efficient heat sink due to the Silicon substrate. However, the heat spreading from the diode to the silicon substrate is not clear. Please add the clarification of the heat spreading mechanisms and potential limitations.

Response: We performed simulation of the heat dissipation in our device. The following statements have been added to Supplementary Section 6.

Supplementary Section 6.

“We performed simulation of the heat dissipation in our device. In Fig. 1 (a), we present the simulated structure of our device where we use homogeneous SiO₂, poly-Si and crystalline Si to approximate the back-end-of-line dielectrics, the n-poly-Si contact, and the substrate, respectively. The heat transfer equation is solved in 3-D assuming a cylindrical device geometry. We assume the bottom boundary of the substrate is fixed at 300 K and all the other boundaries are thermally isolated.

At the interface of the poly-Si and the c-Si, a 2-dimensional heat source with 300 nm diameter and 13.5 mW power is set. We estimate an upper bound for heat generation by assuming that the voltage across the device - excluding the n-poly-Si contact - is all on the Si filament and the heat exchange from the carriers to the lattice happens all in the active region. Specifically, at 6 mA, the measured total voltage is 5.9 V, the calculated voltage drop on the n-poly-Si contact is 3.6 V based on its geometry and sheet resistance, and hence the estimated voltage across the nanoscale emitter is 2.3 V. We also assume the active region is the same size as the emission spot (0.09 +/- 0.04 μm² at 6 mA).

The simulated temperature distribution in the proximity of the heat source is presented in Fig. 1 (b). The temperature spatially decreases to room temperature within 1 μm from the heat source and the maximum local temperature rise is approximately 170°C. Note that this is the upper limit of temperature rise in our device, which justifies that the substrate is an efficient heat sink.

As a comparison, we simulated an alternative device structure where SiO₂ is used to confine the current flow instead of a filament through the top oxide. In Fig. 1 (c) we use the identical geometry of Fig. 1 (a) but with a 0.3 μm thick SiO₂ layer around the heat source. This represents a design which has an active region of the same size of Fig. 1 (a) but the carriers are confined by STI instead of an electrical field. The simulated temperature distribution is presented in Fig. 1 (d). Here the maximum local temperature rise is approximately 440°C, which is more than twice of the temperature rise in Fig. 1 (b). We further increase the thickness of the STI to 0.5 μm and 1 μm for stronger carrier confinement, and the maximum local

temperatures are approximately 550°C and 780°C, respectively. Strong Auger recombination and irreversible device degradation are likely to happen at these high temperatures.

The results above clearly show that efficient heat dissipation from the active region to a heat sink is required to ensure moderate local temperature rise. In our device, this is achieved by confining carriers using a local electrical field while leaving the heat conduction path to the substrate unaffected. The main limitation of the current design is that even though the lattice temperature rise is small the hot electrons injected from the n-poly-Si can still damage the device. This is likely the cause of the irreversible degradation at higher injection. We can further optimize the n-contact and fabricate shallow junctions near the active region to improve the reliability.”

In the main text, we added a sentence in the paragraph of the emission mechanism to point the readers to the supplementary information. (line 278)

“...This is elaborated in Supplementary Section 6 where the heat dissipation from the active region to the substrate is simulated and compared with alternative device configurations...”

Fig. 1. Simulation results of heat dissipation. (a) Simplified structure of our device assuming rotational symmetry around z axis. (b) Stationary temperature distribution of (a) near the heat source. (c) Device structure similar to (a) but with STI confinement. (d) Stationary temperature distribution of (c) near the heat source. The simulation was performed in COMSOL Multiphysics 5.3a. The thermal conductivities of c-Si, poly-Si, and SiO_2 are $156 \text{ W}/(\text{m}\cdot\text{K})$, $30 \text{ W}/(\text{m}\cdot\text{K})$, and $1.3 \text{ W}/(\text{m}\cdot\text{K})$, respectively. [Physical review 134.4A (1964): A1058] [Journal of Microelectromechanical Systems 10.3 (2001): 360-369.] [Fundamentals of semiconductor processing technology. Kluwer Academic Publishers, 1995]

2) The authors should specify the limits on current injection and highest output power that is possible to extract out of the LED.

Response: The maximum extracted power is approximately 94.0 pW in free space and 42.1 pW in a single-mode fiber at 6 mA injection. We were able to inject 7 mA current but the device became unreliable. The optical power decreased more than 50% within a minute.

We have added the measured limit at the end of the characterization the discussion section. (line 521)

“...The maximum optical powers extracted from a single LED are approximately 94 pW in free space and 42 pW in a single-mode fiber at 6 mA injection...”

The limit of current injection is mentioned when we discuss the reliability of the device. (line 432)

“...The device becomes less reliable at high injection. With 7 mA injection, the optical power decreases more than 50% within a minute...”

3) Unless I overlooked it, the manuscript does not specify at which temperature were the measurements carried out. Furthermore, it would be highly complementary to measure the light emission/EQE at different temperatures. The authors must consider additional measurements at different temperatures to have additional insights on the physical mechanisms limiting efficiency.

Response: The measurements in the main text were all carried out at room temperature ($\approx 22^\circ\text{C}$). We have added this statement in the introduction section (line 141):

“In this work, we report an electrically-driven, nanoscale Si light-emitting diode (LED) realized in an unmodified, open-foundry microelectronic CMOS node. At room temperature $\approx 22^\circ\text{C}$, our LED exhibits sub-wavelength emission area...”

and line 403:

“...The results presented above are all at room temperature and at steady state. In Supplementary Section 7, we present the device performance at elevated temperatures. We observe that the SMF-coupled power increases with temperature from 10°C to 70°C ...”

We performed additional measurements on the SMF-coupled power with the chip die mounted on a temperature-controlled heat sink. The results and discussions are added to Supplementary Section 7.

Supplementary Section 7.

“We performed measurements on the SMF-coupled power with the chip die mounted on a temperature-controlled heat sink. (Fig. 2) We observe that the SMF-coupled power increases monotonically with temperature from 10°C to 70°C . A similar trend of emission enhancement has been reported by Ng et al. [(Nature 410.6825 (2001): 192-194.)] where the emission increases with temperature from 80 K to approximately room temperature. This trend suggests that Auger recombination, which is usually the main cause of thermal droop, does not dominate the carrier recombination in the active region.

Multiple factors can contribute to emission enhancement with temperature. First, at the same current density, the electron concentration near the filament can increase with temperature because the electron mobility and the electron drift velocity in the filament *decrease* with temperature. Second, the hole concentration in the accumulation layer can also increase with temperature. The hole accumulation layer in our device is analogous to the inversion layer of a p-MOSFET of which the threshold voltage *decreases* with temperature. [IEEE transactions on Electron Devices 18.6 (1971): 386-388] If we assume the gate voltage (voltage between the n-poly-Si and the n-well in our device) and the gate capacitance do not change with temperature, the hole concentration increases. Moreover, as pointed out by Ng et al., the

scattering rate of the confined carriers and the effective density of states increase with temperature, which can contribute to the increasing emission power.

Even though the device favors an elevated temperature near room temperature, we emphasize that heat dissipation is essential because the emission enhancement is based on the mitigation of Auger recombination which increases significantly with temperature. Moreover, we notice that at 85°C the optical power starts to decrease over time with 6 mA current, while the device is reliable with this current at 25°C. This suggests that breakdown site propagation, as a thermal runaway process, may happen with lower current at higher temperature.”

Fig. 2. Normalized SMF-coupled power at multiple temperatures. The data points are normalized by the power with 6 mA at 25°C. The inset figure is the SMF-coupled power at 6 mA versus the heat sink temperature.

4) The authors should mention the potential modifications of the structure to achieve higher optical output powers.

Response: We have added the potential modification in the Discussion Section. (line 534)

“The device performance can potentially be improved by several modifications. First, the top contact should be optimized to support higher electron injection given that both the EQE and the optical power increase with current. This can be achieved, for example, by replacing sharp corners of the poly-Si contact by rounded ones to lower local electrical fields and prevent lateral propagation of breakdown sites. It is also possible to silicide part of the contact taper to reduce the parasitic voltage drop. Second, shallow junctions can be fabricated close to the active region to provide stronger electron confinement. Moreover,

the p-substrate contact, which is approximately 2 mm away from the active region, can be fabricated closer to the n contact. This can lower the background emission and increase the bandwidth.”

5) The authors mentioned that the LED structures on multiple chips do not vary significantly. This variance should be quantified to give insights on the reproducibility of the fabrication process (especially the reproducibility of the silicon filament formation is a concern to me).

Response: We have quantified the variance of the device performance in Supplementary Section 8.

Supplementary Section 8.

“In Fig. 3, the SMF-coupled powers and the forward bias voltages of five LEDs on different chip die are presented. These curves were measured after gate oxide breakdown and at room temperature. The relative standard deviations are presented in the inset figures. At 6 mA injection, the standard deviations of the SMF-coupled power and the bias voltage are approximately 5.0 mW and 0.34 V, respectively, which are $\approx 13.7\%$ and $\approx 5.3\%$ of the mean values. These preliminary results indicate good reproducibility of our devices.

The good reproducibility of the silicon filament formation has also been reported in the literature. For example, these Si filaments (usually referred as anti-fuses in analog circuit communities) can be arrayed precisely in standard CMOS platforms as one-time programmable read-only memory (OTP-ROM). [IEEE Electron Device Letters, 24(9), 589-591.] [2009 IEEE International Conference of Electron Devices and Solid-State Circuits (EDSSC), pp. 457-460, 2009.]

The variance of our device is probably due to the spatial randomness of the breakdown site within the contact area of the n-well and the gate oxide ($\approx 0.3 \mu\text{m}^2$). For example, if the breakdown happens near the interface of the n-well and the STI, a current leakage path may form. This is likely the situation of LED1 with the lowest SMF-coupled power since it also has the lowest forward bias voltage. We expect the reproducibility will be further improved with optimized design of the shape of the poly-Si contact.

...”

The results are also mentioned in the paragraph discussing the performance variance. (line 415)

“The performance variance of our LEDs on multiple chip die is modest. In Supplementary Section 8, we present the SMF-coupled powers and the bias voltages of five devices on different chip die. The maximum relative standard deviations of the SMF-coupled powers and the bias voltages are 13.7% ($\sigma_{Power,6mA} \approx 5.0 \text{ mW}$) and 5.3% ($\sigma_{Voltage,6mA} \approx 0.34 \text{ V}$), respectively. The power of the worst device is approximately 80% of the normal devices. These preliminary results indicate good reproducibility of our devices...”

Fig. 3. Variance of device performance. (a) Single-mode fiber (SMF) coupled power and (b) bias voltage versus current in five LEDs on different chip die after gate oxide breakdown.

To Reviewer 2:

Nanoscale on-chip light sources with high intensity are desired for various applications in integrated photonics systems. This paper reported an electrically driven Si light-emitting diode (LED) with sub-wavelength emission area fabricated in an open-foundry microelectronics CMOS platform. The emission area is smaller than $0.14 \mu\text{m}^2$ and this LED has high spatial intensity of $> 50 \text{ mW/cm}^2$, while the LED emission spectrum is centered around 1100 nm. This work is very impressive and it can be accepted if the authors address the following comments.

1. The authors mentioned “In our previous work, we have shown that vertical pn junctions with top metal contact can support high injection current while also lower the device footprint compared to lateral junctions.³² However, the opaque metal contact leads to significant shadowing if the emission area shrinks to a comparable size. ... Compared with the work on the smallest CMOS emitters,³² the emission area is 2 orders of magnitude smaller and the average intensity is approximately doubled”. From this description, the difference between the present work and the previous work in 32 is that the top contact is replaced by a n-poly-Si. Is there any other significant difference? Please clarify more details about this. What is the major reason for the 2 orders of magnitude improvement?

Response: Although both devices are based on vertical junctions, the LED presented in this report and the previously published LED are significantly different in three aspects:

1. The top contact of the presented LED is fabricated using poly-Si instead of metal, which prevents shadowing and thus enhances light extraction. This is especially essential when the emission area is nanoscale because standard CMOS processes typically do not allow metal contact smaller than 100 nm.
2. In the presented LED, the top contact and the active region are not directly in contact, but through the sub-100 nm Si filament formed in the gate oxide. This configuration spatially confines holes in all x-, y-, and z-directions, which reduces the emission area and enhances the spatial intensity. In contrast, in the previously published LED, the top contact is directly fabricated on the bulk active region where no significant carrier confinement is achieved. The carriers can diffuse several to tens of micrometers laterally and thus the emission area is orders of magnitude larger than the presented LED.
3. In the presented LED, the active region is away from surfaces except the well-passivated gate oxide. The SRH recombination is thus minimized. In the previously published LED, the optimal structure contains multiple interfaces in the active region including metal/p+, p+/p-well, p-well/p-, p-/n-well, n-well/substrate, as well as the interfaces with STI. Therefore, a higher SRH recombination rate is expected.

These three main differences together lead to the enhancement of the spatial intensity and the much smaller emission area of the present LED in this report.

We have added the clarification in the main text. (line 358)

“...Compared with the smallest CMOS emitters reported previously [IEEE Transactions on Electron Devices 68.8 (2021): 3870-3875.], the emission area is 2 orders of magnitude smaller and the average intensity is approximately doubled. The improvement of our LED is achieved first by using transparent top contact to prevent shadowing and enhance light extraction. Also, the top contact and the active

region are not directly in contact, but through the sub-100 nm Si filament formed in the gate oxide. This configuration spatially confines holes in all x-, y-, and z-directions, which reduces the emission area and enhances the spatial intensity. In contrast, in [IEEE Transactions on Electron Devices 68.8 (2021): 3870-3875.], the top contact is directly fabricated on the bulk active region and the carriers can diffuse several to tens of micrometers laterally. Moreover, SRH recombination is minimized because the active region is away from surfaces except the well-passivated gate oxide while in [IEEE Transactions on Electron Devices 68.8 (2021): 3870-3875.] multiple interfaces exist in the active region and a higher SRH recombination rate is therefore expected.”

2. The present LED emits with the emission spectrum centered around 1100 nm. Please tell a little more details about the potentials the present LED for various applications.

Response: We have added the potential application in the end of Discussion Section. (line 564)

“Besides the demonstrated application in holography, the presented LED is potentially useful in multiple other scenarios. For example, since the wavelength is within the minimum absorption window of biological tissues [Nature nanotechnology 4.11 (2009): 710-711.], together with its high intensity and nanoscale emission area, the LED can be ideal for bio-imaging and bio-sensing applications, including near-field microscopy and implantable CMOS devices. Also, it is possible to integrate the LED with on-chip photodetectors and the LED can then find its applications in inter-chip communication, NIR proximity sensing, and on-wafer testing of photonics.”

3. Is the wavelength tunable?

Response: The LED itself is not wavelength-tunable. However, given its broadband spontaneous emission spectrum (> 150 nm FWHM), it is possible to filter or enhance the emission by coupling the LED with additional resonant structures. For example, on-chip micro-rings can be used to tune the spectral transmission by electro-optic effects. [Nature 435.7040 (2005): 325-327.]

4. The authors described the formation of the Si filament whose size is about usually sub-100 nm. Is the position of the Si filament controllable, or is it a kind of random? Is there any influence on the device performance?

Response: The position of the Si filament is controllable. These Si filaments (usually referred as anti-fuses in analog circuit communities) can be arrayed precisely in standard CMOS platforms as one-time programmable read-only memory (OTP-ROM). [IEEE Electron Device Letters, 24(9), 589-591.] [2009 IEEE International Conference of Electron Devices and Solid-State Circuits (EDSSC), pp. 457-460, 2009.]

In our work, the position of the Si filament is well controlled within the area where the gate oxide is in contact with both the poly-Si and the n-well. In the current design, this area is defined by STI and is approximately 500nm X 500nm. If higher spatial precision is desired, the contact area can be further decreased down to the minimum feature size of the CMOS node (55 nm in this report). It is also possible to fabricate geometrical features and change doping profiles in the poly-Si layer to generate a strong

local electrical field and breakdown the gate oxide, which may have even higher precision. However, such a high field might introduce unwanted defects and leakage current.

The position of the Si filament may have influence on the performance. For example, as mentioned above, if the breakdown point is on the edge of the poly-Si contact, there may be leakage current through defects. However, according to the test results on multiple devices (Supplementary information: variance and reliability), the variance of the performance is modest. The optical power of the worst device is approximately 80% of the normal devices. We expect the variance will further decrease with optimized design of the shape of the poly-Si contact.

We have quantified the variance of the device performance in Supplementary Section 8.

Supplementary Section 8.

“In Fig. 4, the SMF-coupled powers and the forward bias voltages of five LEDs on different chip die are presented. These curves were measured after gate oxide breakdown and at room temperature. The relative standard deviations are presented in the inset figures. At 6 mA injection, the standard deviations of the SMF-coupled power and the bias voltage are approximately 5.0 mW and 0.34 V, respectively, which are $\approx 13.7\%$ and $\approx 5.3\%$ of the mean values. These preliminary results indicate good reproducibility of our devices.

The good reproducibility of the silicon filament formation has also been reported in the literature. For example, these Si filaments (usually referred as anti-fuses in analog circuit communities) can be arrayed precisely in standard CMOS platforms as one-time programmable read-only memory (OTP-ROM). [IEEE Electron Device Letters, 24(9), 589-591.] [2009 IEEE International Conference of Electron Devices and Solid-State Circuits (EDSSC), pp. 457-460, 2009.]

The variance of our device is probably due to the spatial randomness of the breakdown site within the contact area of the n-well and the gate oxide ($\approx 0.3 \mu\text{m}^2$). For example, if the breakdown happens near the interface of the n-well and the STI, a current leakage path may form. This is likely the situation of LED1 with the lowest SMF-coupled power since it also has the lowest forward bias voltage. We expect the reproducibility will be further improved with optimized design of the shape of the poly-Si contact.

...”

The results are also mentioned in the paragraph discussing the performance variance. (line 415)

“The performance variance of our LEDs on multiple chip die is modest. In Supplementary Section 8, we present the SMF-coupled powers and the bias voltages of five devices on different chip die. The maximum relative standard deviations of the SMF-coupled powers and the bias voltages are 13.7% ($\sigma_{Power,6mA} \approx 5.0 \text{ mW}$) and 5.3% ($\sigma_{Voltage,6mA} \approx 0.34 \text{ V}$), respectively. The power of the worst device is approximately 80% of the normal devices. These preliminary results indicate good reproducibility of our devices...”

Fig. 4. Variance of device performance. (a) Single-mode fiber (SMF) coupled power and (b) bias voltage versus current in five LEDs on different chip die after gate oxide breakdown.

5. Could this micro-LED be modulated fast? It will be very nice if the authors can give a comment and explanation on this.

Response: We have measured the time-resolved optical power under voltage pulses using time-correlated single photon counting (TCSPC) and the frequency-dependent emission power under square waves. (Please see Fig. 5.) In short, the 3-dB switching bandwidth is approximately 77 MHz under 0 – 4V, 50% duty cycle square waves. The bandwidth is one order of magnitude higher than the reported fastest forward biased CMOS emitter. [IEEE Transactions on Electron Devices 65.10 (2018): 4413-4420.] The fast modulation is a result of the small active region (essentially only the hole accumulation layer and the Si

filament). The bandwidth of our device is mainly limited by the diffusion capacitance of the substrate since our current design is not optimized for high-speed hole injection. Specifically, the contact pad to the substrate is approximately 2 mm away from the active region, which significantly increases the diffusion capacitance. This can be easily modified in future designs.

Detailed discussions are added to the Supplementary Section 9.

Supplementary Section 9.

“We used time-correlated single-photon counting (TCSPC) technique to measure the time-resolved SMF-coupled optical power under rectangular voltage pulse. An arbitrary wave generator (AWG, HP) was used to bias the LED and a silicon single photon avalanche diode (SPAD, Perkin Elmer), of which the quantum efficiency (QE) is > 10% at 1 μm , was used to detect optical pulses. The electronic signals were processed using a TCSPC timing module (B&H).

The TCSPC results are presented in Fig. 5 (a, b). The optical pulses have distinct rise and fall edges. In Fig. 5 (b), a 250 ns voltage pulse is applied and the LED reaches its steady state. The 10% - 90% rise and fall time are approximately 42.3 ns and 3.5 ns, respectively. The relatively long rise time is likely due to the diffusion capacitance of the substrate. In our current design, the contact to the p-substrate is approximately 2 mm away from the active region, which is not optimized for high-speed hole injection. Meanwhile, the fall time is relatively short because the holes can be swept out from the n-well by the built-in electrical field [Light-Emitting Diodes, Cambridge Univ. Press, 2006]. Similar carrier dynamics with asymmetric rise and fall edges have been reported in the literature. For example, in [Nature 435.7040 (2005): 325-327.], Xu et al. report a Si electro-optic modulator based on a p-i-n ring resonator of which the carrier injection time (electrical rise time) is approximately 10 ns while the carrier extraction time (electrical fall time) is 100s of ps. In [IEEE Transactions on Electron Devices 65.10 (2018): 4413-4420.], Puliyanokot et al. report fast-switching, forward-bias p-i-n LEDs fabricated in SOI process. The bandwidth (10 MHz) of their fastest device is limited by the relatively long rise time while the fall time can be neglected.

Although the rise time to the steady state is relatively long, the time from 10% to 50% is only 4.7 ns, which is comparable to the fall time. The rising edge of the optical power corresponds to a 3-dB switching bandwidth on the order of 100 MHz. In Fig. 5 (c), we present the DC optical power when the LED was modulated by square waves with 50% duty cycle and 50% DC offset. The DC component of the optical power was measured by a low bandwidth (< 20 Hz) photodiode (HP). Since the fall time is shorter than the rise time, the DC optical power is mainly determined by the rising edge. With 0 – 4 V and 0 – 3 V voltage swing, the 3-dB bandwidths are 77 MHz and 51 MHz, respectively. These bandwidths are one order of magnitude higher than those reported in [IEEE Transactions on Electron Devices 65.10 (2018): 4413-4420.]. The fast modulation is a result of the small active region and we expect higher bandwidths from future designs since our current device is not optimized for high speed modulation.”

Fig. 5. Time-resolved optical power and modulation speed of our LED. (a, b) Time-resolved SMF-coupled optical power under rectangular voltage pulse. The pulse is 0 - 4 V and the pulse widths are (a) 5 ns, 10 ns, 50 ns and (b) 250 ns. In (b), the 10%, 50% and 90% power levels are indicated by the dashed black lines. The 10% - 90% rise and fall time are also labeled. (c) SMF-coupled DC optical power with the bias modulated by 50% duty cycle square waves. The voltage swings are 0 - 4 V and 0 - 3 V.

REVIEWERS' COMMENTS

Reviewer #1 (Remarks to the Author):

The authors responded appropriately to my concerns.

Reviewer #2 (Remarks to the Author):

The authors have addressed all the comments, and thus this paper could be accepted.